# Targeting Histone Deacetylases to Modulate Graft-Versus-Host Disease and Graft-Versus-Leukemia

**DOI:** 10.3390/ijms21124281

**Published:** 2020-06-16

**Authors:** Sena Kim, Srikanth Santhanam, Sora Lim, Jaebok Choi

**Affiliations:** 1Division of Oncology, Department of Medicine, Washington University School of Medicine, St. Louis, MO 63110, USA; slim22@wustl.edu; 2Independent Researcher, St. Louis, MO 63122, USA; sree_canth@yahoo.com

**Keywords:** histone deacetylase inhibitor, graft-versus-host disease, graft-versus-leukemia

## Abstract

Allogeneic hematopoietic stem cell transplantation (allo-HSCT) is the main therapeutic strategy for patients with both malignant and nonmalignant disorders. The therapeutic benefits of allo-HSCT in malignant disorders are primarily derived from the graft-versus-leukemia (GvL) effect, in which T cells in the donor graft recognize and eradicate residual malignant cells. However, the same donor T cells can also recognize normal host tissues as foreign, leading to the development of graft-versus-host disease (GvHD), which is difficult to separate from GvL and is the most frequent and serious complication following allo-HSCT. Inhibition of donor T cell toxicity helps in reducing GvHD but also restricts GvL activity. Therefore, developing a novel therapeutic strategy that selectively suppresses GvHD without affecting GvL is essential. Recent studies have shown that inhibition of histone deacetylases (HDACs) not only inhibits the growth of tumor cells but also regulates the cytotoxic activity of T cells. Here, we compile the known therapeutic potential of HDAC inhibitors in preventing several stages of GvHD pathogenesis. Furthermore, we will also review the current clinical features of HDAC inhibitors in preventing and treating GvHD as well as maintaining GvL.

## 1. Introduction

Allogeneic hematopoietic stem cell transplantation (allo-HSCT) is the only curative therapy for various malignant and nonmalignant hematopoietic disorders including bone marrow failure states, hemoglobinopathies, and thalassemia [1,2]. The therapeutic benefit of allo-HSCT in malignant disorders relies on strong alloimmune responses between donor T cells and antigen-presenting cells (APCs), including recipient hematopoietic APCs, recipient malignant cells (i.e., B cell lymphoma), and donor bone marrow-derived APCs [3,4,5,6,7,8]. The alloimmune response can eliminate residual malignant cells resulting in a reduced chance of relapse. This anti-tumor effect is known as the graft-versus-leukemia (GvL) effect. Unfortunately, GvL is accompanied by detrimental graft-versus-host disease (GvHD), which is the main complication after allo-HSCT [9,10]. The standard prophylactic treatments with cyclosporine, tacrolimus (FK506), and sirolimus globally target immune cells and impair their normal immune functions, which in turn results in suppressed GvHD. However, these treatments also increase the risk of leukemia relapse due to the impaired donor T cell activity [11,12,13]. Therefore, alternative and more selective therapeutic strategies are needed to suppress GvHD without affecting GvL. 

Acetylation is one of the post-translational modifications that can alter gene expression and protein function [14,15]. It is regulated by histone deacetylases (HDACs) and their counterpart histone acetyl-transferases (HATs). HDACs remove the acetyl groups which are added by HATs in histones and non-histones [16]. The acetylation status regulated by HDACs and HATs affects epigenetic changes associated with tumorigenesis and aberrant inflammatory responses [17,18,19]. Dysfunction of HDACs and HATs by loss or gain-of-function mutations, abnormal recruitment to gene promoters, and aberrant expression has been found in hematologic (acute myeloid leukemia, acute and chronic lymphocytic leukemia, diffuse large B-cell lymphoma, cutaneous T-cell lymphoma, and Hodgkin lymphoma) and solid malignancies (neuroblastoma, medulloblastoma, lung, gastric, liver, pancreatic, colorectal, breast, ovarian, prostate, renal, bladder, melanoma, oral, endometrial, pancreatic, thyroid, and esophageal cancer) [19,20,21,22,23,24,25]. Indeed, several HDAC inhibitors have been shown to play as suppressors of tumorigenesis and angiogenesis, upregulating growth-suppressive proteins such as p21, survivin, and transforming growth factor-beta (TGF-β) receptor [26,27]. In addition, the altered status of HDAC activity and expression have been found in numerous inflammatory diseases such as arthritis, inflammatory bowel diseases, septic shock, ischemia-reperfusion injury, airways inflammation and asthma, diabetes, age-related macular degeneration, cardiovascular diseases, and multiple sclerosis [28,29,30]. Many studies suggest that the anti-tumor and immunomodulatory effect by HDAC inhibition can be used against cancer and GvHD after allo-HSCT. In this review, we focus on the therapeutic potential of HDAC inhibitors on each stage of GvHD pathogenesis and GvL after allo-HSCT. Furthermore, we will briefly review the current knowledge on clinical observations of HDAC inhibitors in preventing and treating GvHD while maintaining GvL.

## 2. Classification of HDACs

HDACs are a class of enzymes that deacetylate the acetyl-lysine residues in histones and various nonhistone proteins. In mammalian cells, 18 HDACs have been reported and are divided into four classes: class I (HDACs 1, 2, 3, 8), class II (HDACs 4, 5, 6, 7, 9, 10), class III (sirtuin family: SIRT1-SIRT7), and class IV (HDAC11) based on phylogenetic and sequence similarity to yeast deacetylases (Table 1). Class I HDACs are homologous to yeast RPD3. Class II HDACs are homologous to yeast HDA1 and larger than other class HDACs. The class II HDACs can be subdivided into class IIa (HDACs 4, 5, 7, 9) and class IIb (HDACs 6, 10), depending on the double catalytic domain. HDAC11 only belongs to class IV HDAC and is the smallest classic HDAC. Class III HDACs are homologous to yeast SIR2. Classes I, II, and IV HDACs are classic HDACs that require zinc ions to deacetylate their substrates and can be inhibited by classic HDAC inhibitors through a conserved functional catalytic domain. Class III sirtuins require nicotinamide adenine dinucleotide as an essential catalytic cofactor [31,32,33,34].

## 3. Pathogenesis of Acute Graft-Versus-Host Disease

Before allo-HSCT, patients receive conditioning regimens that involve cytotoxic chemotherapy and/or total body irradiation to allow efficient donor stem cell engraftment and prevent rejection of the graft by killing residual cancer cells and suppressing the recipient’s immune system [35,36]. However, it also damages host intestinal mucosa and other tissues resulting in the initiation of GvHD development [37,38]. The pathophysiology of GvHD happens in several phases with the initial phase beginning due to the damaged intestinal epithelium by the conditioning regimens (Figure 1a). The loose intestinal integrity after conditioning allows the release of intestinal microbes, pathogen-associated molecular patterns (PAMPs), damage-associated molecular patterns (DAMPs), and pro-inflammatory cytokines to cross the epithelial barrier and enter the blood circulation [39,40,41]. Subsequently, translocation of these pathogenic components activates APCs by triggering toll-like receptor (TLR) signaling pathways (Figure 1b) [42,43,44]. The signaling cascade activates both recipient APCs and donor hematopoietic cell-derived APCs by promoting protein expression of major histocompatibility complex (MHC) I, MHC II, and co-stimulatory molecules such as CD80, CD86, and CD40, which are associated with T cell allogeneic activity [45,46,47,48]. During their activation, APCs take up antigens through receptor-mediated uptake, phagocytosis, and pinocytosis. The processed antigens are presented to donor T cells by recipient hematopoietic APCs at an early time point and cross-presented by donor hematopoietic cell-derived APCs at a later time point when donor hematopoietic cells have fully replaced host hematopoietic APCs. The recipient antigen presented on both hematopoietic recipient and donor-derived APCs has a positive impact on GvHD and GvL. Interestingly, alloantigens expressed on non-hematopoietic APCs such as epithelial cells have been shown to exacerbate acute GvHD while reducing the GvL effect [49,50,51,52]. After recognition of the alloantigens and co-stimulatory molecules through T cell receptor (TCR) and CD28 on donor T cells, the donor T cells are rapidly activated and expanded. These T cells differentiate into effector T cells such as T helper (Th) 1, Th2, Th17, and induced regulatory T (Treg) cells (Figure 1c) [53,54]. In the last phase of GvHD pathogenesis, alloantigen-stimulated T cells traffic into GvHD target organs such as the gastrointestinal (GI) tract, skin, lung, and liver (Figure 1d) and cause tissue damage (Figure 1e). Figure 1 summarizes GvHD pathogenesis. Several groups have demonstrated that these detrimental effects of donor T cells can be separated from the beneficial GvL using HDAC inhibitors. Considering the anti-tumor and immunomodulatory properties of HDAC inhibitors as discussed below, targeting HDACs may be a promising strategy to prevent GvHD while preserving the GvL effect. 

## 4. Therapeutic Potential of HDAC Inhibitors in GvHD

### 4.1. Effect of HDAC Inhibitors on Each Stage of GvHD Pathogenesis

#### 4.1.1. Effect of HDAC Inhibitors on the Intestinal Barrier Damaged by the Conditioning Regimen

Studies have shown that the release of pathogenic components by increased gut permeability or damaged cells is considered as an initial stage for GvHD pathogenesis. Butyrate, a short-chain fatty acid produced by the colon microbiota, is a well-known pan-HDAC inhibitor [55,56]. Butyrate promotes the intestinal epithelial barrier function by promoting redistribution of tight junction proteins during calcium switch-induced tight junction assembly and increasing transepithelial electrical resistance in human colonic Caco-2 and T84 cell monolayer models [57,58]. In addition, sodium butyrate reduces cecal ligation and puncture (CLP)-induced mortality through the protection of intestinal barrier as measured by fluorescence-labeled macro-molecule FD40 in plasma [59]. Moreover, in vivo administration of pan-HDAC inhibitors, givinostat and vorinostat (also known as suberanilohydroxamic acid, SAHA), results in an improved barrier recovery and epithelial wound healing in a dextran sodium sulfate (DSS)-induced mouse model of inflammatory bowel disease [60]. These regenerative effects are derived from increased secretion of TGF-β, IL-8, and expression of the tight junction proteins claudin-1/2 and occludin. Treatment with valproic acid (VPA), another pan-HDAC inhibitor, stabilizes the intestinal claudin-3 that is essential for the formation and maintenance of mucosal tight junction integrity and suppresses the leakage of harmful pathogenic components from the intestinal lumen into systemic circulation in hemorrhagic shock (HS) [56,61]. In addition, selective HDAC6 inhibitors, ACY-1083 and tubastatin-A, attenuate intestinal inflammation and preserve intestinal tight junction integrity through suppression of neutrophil infiltration, pro-inflammatory cytokines, and apoptosis in a rat model of HS [56]. Santacruzamate A is a selective HDAC2 inhibitor. Treatment with Santacruzamate A shows the protection of the intestinal mucosal barrier with increased expression of ZO-1 and occludin in a rat model of galactosamine/LPS-induced acute liver failure (ALF) [62]. Moreover, the treatment with Trichostatin A (TSA), a prototypical pan-HDAC inhibitor, dramatically improves intestinal permeability in the ALF rat model [63]. Entinostat inhibits class I HDACs and improves experimental cholera through the restoration of epithelial barrier integrity in a rabbit cholera model [64]. Furthermore, it has been shown that caprylic acid and nonanoic acid suppress bacterial translocation across the intestinal cell line, IPEC-J2, while upregulating endogenous host defense peptides, beta-defensin1/2, through an increased intestinal epithelial immunological barrier function by inhibition of HDAC activity [65]. These results suggest that inhibition of HDAC can attenuate inflammatory responses triggered by pathogenic components by promoting the recovery of intestinal barrier function that is damaged by conditioning regimens.

#### 4.1.2. Effects of HDAC Inhibitors on Cytokine Production and APCs Activation

The intestinal microbial and host components are recognized by TLRs and activate downstream cascades via recruitment of two essential cytosolic TLR transducers, MYD88 and TRIF. MYD88/TRIF activate MAPKs and transcription factors such as NF-κB, activator protein 1 (AP-1), and interferon regulatory factor 3/7 (IRF-3/7) to induce expression of pro-inflammatory cytokines, chemokines, and type I IFNs [66,67]. MYD88 can be directly deacetylated by the cytoplasmic HDAC6, and its hypoacetylation enhances pro-inflammatory cytokine production [68,69]. Several studies have suggested that the TLR signaling pathway is responsible for the development of pathogenic alloreactive T cells by activation of APCs with increased alloantigen presentation, pro-inflammatory cytokine production, and protein expression of MHC I/II and co-stimulatory molecules [42,44,70,71].

Numerous studies have demonstrated that TLR signaling pathways are blocked by HDAC inhibitors in diverse cell types and experimental animal disease models. In genome-wide microarray analysis, TSA inhibits the expression of innate immune genes that are up-regulated by pattern recognition molecules, LPS (TLR4 agonist) or Pam3CSK4 (TLR1/2 agonist), in the bone marrow-derived macrophages [72]. The genes whose expression is inhibited by TSA are involved in microbial sensing and killing, inflammatory cytokine production, cell cycle regulation, apoptosis, and antigen processing and presentation: *Tlrs*, *Cd14*, *Md-2*, *Aim1*, *Nlrp3*, *Nod1/2*, *Pycard/Asc*, *integrins*, and so on. Moreover, TSA and vorinostat interfere with transcription of LPS-induced pro-inflammatory cytokines (IL-12p40, TNF-α, IL-6, IL-1β, and IFN-β) and costimulatory molecules (CD86 and CD40) by blocking the recruitment of transcription factors to target promoter regions in macrophages and DCs. Treatment with VPA also reduces the production of IL-12 and TNF-α as well as expression of CD40, CD86, and CD80 through the change of macrophage phenotype into anti-inflammatory M2 in the LPS-induced mouse macrophage cell line RAW264.7 and primary mouse bone marrow macrophages [73]. Apicidin also inhibits the expression of MHC I/II, CD80, and CD86 in bone marrow-derived dendritic cells (BMDCs) by inhibiting LPS-induced HDAC activity [74]. RGFP966, a selective HDAC3 inhibitor, inhibits LPS-induced microglia activation via the reduction of LPS-induced expressions of TLRs, CD36, spleen tyrosine kinase, and pro-inflammatory cytokines TNF-a and IL-6 [75]. Another HDAC 1–2 inhibitor, KBH-A42, reduces LPS-induced endotoxemia [76]. Administration of ACY-1251, a selective HDAC6 inhibitor, significantly decreases the protein expression of TLR4 and activation of MAPKs (JNK, p-38, and p-ERK) and NF-κB signaling in a mouse model of LPS-induced ALF [77]. In a cigarette smoke-exposed lung inflammation mouse model, Entinostat enhances anti-inflammatory cytokine IL-10, resulting in attenuation of the expression of pro-inflammatory cytokines and a decrease of neutrophil influx into the lungs [78]. In a rat seizure behavior model, vorinostat suppresses kainic acid-induced microglia activation and neuron apoptosis through the inhibition of TLR4, MYD88, NF-κB component p65, and IL-1β mRNA and protein expression in the hippocampus [79]. Vorinostat also blocks the expression of MHC II in LPS- and IFNγ-stimulated mesangial cells [80]. In addition, treatment with vorinostat and givinostat ex vivo and in vivo blocks deacetylation of non-histone protein STAT3, which attenuates GvHD by enhancing indoleamine 2,3-dioxygenase (IDO) expression in BMDCs and recipient APCs. During the development of GvHD, up-regulation of IDO expression has been found in GvHD target organs, such as the colon, to suppress inflammatory cytokines in plasma and APCs’ function on allogeneic T cell proliferation and survival [81,82,83,84]. On the other hand, inhibition of class III HDACs SIRT1 increases NF-κB-mediated expression of pro-inflammatory cytokines through the acetylation of p65 subunit of NF-κB [85]. 

These results indicate that inhibition of HDACs can suppress activation of APCs and other immune cells through the reduction of TLR-induced expression of pro-inflammatory cytokines, MHC I/II, and co-stimulatory molecules, whereas inhibition of class III HDACs seems to enhance the TLRs signaling pathway. 

#### 4.1.3. Effect of HDAC Inhibitors on Donor T Cell Activation and Differentiation

Activated APCs present alloantigens with MHC I or II and costimulatory molecules, which are recognized by TCR and CD28, respectively, on T cells. These signals induce T cell activation, proliferation, and differentiation toward effector T cells such as Th1, Th2, and Th17 with the expression of distinct master transcription factors T-bet, GATA-3, and RORγt, respectively [79,86,87]. In addition, cytokines secreted by APCs and other immune cells provide an additional signal to differentiate donor T cells into effector T cells [88]. 

Several HDAC inhibitors have been demonstrated to negatively regulate the activation, proliferation, and differentiation of T cells associated with the development of GvHD. Treatment with TSA and vorinostat has shown to inhibit the capacity of DCs to differentiate naïve T cells into Th1 and Th17 cells, suppressing Th1-attracting chemokines (CXCL9, 10, 11) and Th1- and Th17-inducing cytokines (IL-12 and IL-23, respectively) in LPS/IFN-γ treated DCs [89]. This capacity of DCs can be inhibited by apicidin through the reduced production of IL-12, a Th1-inducing cytokine, by DCs [74]. Dacinostat inhibits the expression of LPS-induced signal for Th1 differentiation, such as costimulatory molecules (CD40 and SLAM), and cytokines (IL-12, IL-15, and EBI3) in human monocyte-derived macrophages and DCs. In addition, dacinostat blocks Th1 activation by suppressing DC-mediated IFN-γ secretion in co-cultures of Th1 cells and BMDCs as well as the production of Th1-attracting chemokines in LPS-treated macrophages and DCs [90]. Butyrate inhibits functional differentiation of monocyte-derived DCs with decreased production of IL-12 p40 and IL-6, resulting in a reduction of Th1 differentiation without affecting the Th2 cells in mixed lymphocyte reaction [91]. However, panobinostat, a hydroxamate-based pan-HDAC inhibitor, accelerates GvHD by upregulating CXCR3 expression on donor T cells and increasing Th1 type cytokines. Although contradictory study results with Th2 cells in GvHD pathogenesis have been reported, inhibition of HDACs shows an unchanged or increased Th2 cell population. Inhibition of SIRT1 by sirtinol results in increased acetylation of GATA3 and thereby reduces its activity as a transcription factor of Th2 cytokine genes [92]. Treatment with vorinostat inhibits Th17 differentiation in mouse models of encephalomyelitis [93], collagen-induced arthritis [94], and experimental autoimmune uveitis [95]. Conditional *Hdac6* KO in CD4+ T cells and two different HDAC6 inhibitors, tubacin and tubastatin-A, reveal increased IL-17 producing cells [96,97], whereas butyrate decreases Th17 cells [98]. Moreover, inhibition of SIRT1 decreases Th17 cell differentiation by hyperacetylating RORγt, thereby reducing transcription of the *IL-17A* gene [99]. In a mouse model of allo-HSCT, VPA attenuates the severity of GvHD by reducing the numbers of Th1 and Th17 cells in the lung and liver at an early stage and decreasing the serum levels of IFN-γ and IL-17 at a late stage [100]. Interestingly, treatment with VPA preserves GvL activity in mice bearing acute myeloid leukemia cells. The immunomodulatory effect of VPA on Th1 and Th17 differentiation was also found in a mouse model of experimental allergic encephalomyelitis [101]. In addition, vorinostat also reduces GvHD without affecting proliferation of donor T cells and cytotoxic function of donor CD8+ cells to alloantigen-expressing cells, thereby resulting in better survival with no leukemia after allo-HSCT [102]. 

Treg cells are negatively associated with the development of GvHD and other inflammatory diseases including arthritis, irritable bowel syndrome, atopic dermatitis, and psoriasis [103,104]. In mouse models of GvHD, both natural and induced Treg cells suppress the proliferation of conventional T cells, thereby preventing GvHD while preserving the GvL effect [105,106]. Additionally, adoptive transfer of Treg cells not only prevents GvHD but also promotes tissue regeneration in the GI tract [107]. Several studies report that the development and function of Treg cells can be regulated by the acetylation status of FOXP3. In addition, hyperacetylation of FOXP3 can prevent the polyubiquitination and proteasomal degradation of FOXP3, leading to an improved stability and transcriptional activity of FOXP3 and GvHD suppression [108,109,110,111,112,113]. 

Consistent with these studies, vorinostat-treated patients demonstrated an increase in the number and function of Treg cells with upregulated CD45R and CD31 that are representative of enhanced suppressive Treg cell function after allo-HSCT [82]. Whereas there is only a limited number of studies that investigated the effect of HDAC inhibitors on Treg cells in allo-HSCT, many studies have been performed outside the field of allo-HSCT, such as solid organ transplant rejection and DSS-induced colitis. Thus, these studies may provide important insights into understanding the mechanisms by which HDAC inhibitors modulate Treg cells. In a mouse cardiac transplant model, conditional *Hdac1* KO in Treg cells resulted in an impaired function of Treg cells and decreased cardiac allograft survival [114]. Conditional KO of *Hdac3* and *Hdac8* in Treg cells leads to large amounts of IL-2 expression and disruption of Treg cell suppressive function, leading to allograft rejection [115,116]. In contrast, conditional *Hdac11 KO* in Treg cells improves long-term survival without the development of arteriosclerosis by increasing Treg cell function in a cardiac transplant mouse model [116,117]. *Hdac6*, 7, *9*, *10* and *sirt1* KO mice exhibit an increase in the expression and transcription activity of FOXP3 as well as the suppressive function of Treg cells [109,110]. Consistent with the genetic deletion of *Hdac6*, treatment with HDAC6-specific inhibitors, tubacin and tubastatin-A, also increases the suppressive function of Treg cells [118]. In addition, the loss of class III HDAC SIRT1 activity in Treg cells improves allograft survival through inhibition of proteasomal degradation of FOXP3 protein, thereby resulting in increased FOXP3 protein and Treg cell suppressive function in a mouse model of cardiac transplant [119]. Comparable results were observed in wild-type allograft recipients treated with SIRT1 inhibitors, EX-527 and splitomicin [119]. Treatment with vorinostat, VPA, and entinostat not only induces the Treg cell phenotype with increased expressions of FOXP3, TGF-β, and glucocorticoid-induced TNF family-related receptor (GITR), but it also promotes the suppressive activity of Treg cells in vivo and in vitro [120]. In the DSS-induced colitis mouse model, treatment with TSA exhibits an increased number of functionally improved Treg cells, which correlates with reduced disease severity. In addition, the modulation of Treg cells by TSA also shows Treg cell-dependent beneficial effects in fully MHC-mismatched cardiac and islet transplant models. Givinostat induces T helper cell differentiation into Treg cells by targeting the IL-6/STAT3/IL-17 pathway, resulting in an improvement in DSS-induced experimental colitis [121]. In addition, treatment with SIRT1 inhibitors, nicotinamide and EX-527, promoted Treg cell development and function in vitro and in vivo [112,119]. In contrast, another study showed that the treatment with TSA in vitro and in vivo enhances the activity of the *Foxp3* promoter while inducing *Foxp3* mRNA decay, resulting in the down-regulation of *Foxp3* expression and reduction of Treg cell development and function [122]. Treatment of mice with entinostat also decreased *Foxp3* expression and suppressive function of Treg cells [122,123]. 

#### 4.1.4. Effect of HDAC Inhibitors on T Cell Trafficking

T cell trafficking is regulated by the expression of specific chemokine receptors on T cells and chemokines released by GvHD target organs and tumor sites [124,125]. It has been proposed that inhibition of HDACs alters the expression of chemokine receptors on T cells and chemokines in GvHD target organs or the tumor environment, resulting in the reduction of T cell trafficking to GvHD target organs while enhancing T cell infiltration into the tumor. For example, loss of HDAC1 impairs the upregulation of CCR4 and CCR6 on T cells, resulting in the abrogation of T cell trafficking towards chemokines CCL17, CCL22, and CCL20 that are highly secreted in the lung [126]. Treatment with TSA and BML-210 restores the expression of the gut-homing receptor, integrin α4β7 and CCR9, on *Batf* KO T cells [127]. CXCR3 is highly up-regulated in Th1-type CD4+ cells and plays an important role in T cell trafficking toward CXCL9-11 expressed in the GvHD target organs such as the liver, intestines, skin, and lungs [128]. Treatment with panobinostat upregulates expression of CXCR3 on T cells and exacerbates GvHD by promoting T cell trafficking to GvHD target organs such as the liver, colon and, intestines [129,130,131]. In contrast, pathological T cell infiltration in the liver is decreased by vorinostat, thereby attenuating GvHD through down-regulation of CXCR3 on donor T cells [131]. Interestingly, CXCL9-11 were also upregulated in the tumor microenvironment, resulting in the promotion of GvL by increasing CXCR3 + T cell infiltration into the tumor. Treatment with Romidepsin, a bicyclic class 1 selective HDAC inhibitor, has been shown to increase the expression of CCL5, CXCL9, and CXCL10 in tumor cells, promoting T cell recruitment into the tumor site and boost T cell-mediated anti-tumor function in multiple lung tumor models [132]. In addition, combination treatment with GSK126 (a methyltransferase inhibitor) and LB201 (a class 1/2 HDAC inhibitor) enhances expression of CXCL9 and 10 in brain tumor cell lines, resulting in increased T cell trafficking toward tumor cells. Thus, regulation of T cell trafficking can be used to modulate GvHD and GvL. However, it has not been successful to selectively inhibit GvHD over GvL using HDAC inhibitors. 

#### 4.1.5. Effect of HDAC Inhibitors on the Destruction of GvHD Target Organs and/or Tumors

The migrated effector T cells to GvHD target organs and/or tumor sites exert their cytotoxic functions through direct cell contact-mediated or cell contact-independent cytokine-mediated cytotoxic pathways, including FAS ligand (CD95L)/FAS and perforin/granzyme pathways [133]. These cytotoxic functions of T cells not only affect activation, differentiation, and recruitment of other immune cells such as macrophages via IFN-γ but also induce cell death of GvHD target organs, tumor cells, and T cells themselves. It has been shown that loss of FAS-FASL signal and perforin-granzyme function by genetic or pharmacologic blockade decreases either GvHD mortality or GvL activity in mouse models of allo-HSCT [134,135,136,137,138,139,140]. 

The cytotoxic function of T cells can be regulated by modifying the expression of cytotoxic molecules using HDAC inhibitors. Treatment with VPA and TSA induces apoptosis of leukemic blasts without affecting normal hematopoietic progenitors through the up-regulation of FAS and FASL expression on leukemic cells [141]. In co-cultures of cytotoxic T cells with glioma cell lines (U251 and GL261), vorinostat and sodium butyrate also enhance FAS/FASL and Perforin/Granzyme B pathway-mediated glioma cell apoptosis [142]. In addition, depsipeptide, a potent histone deacetylase 1/2 inhibitor, enhances effector T cell function by up-regulating Perforin expression in melanoma-specific, antigen-stimulated effector T cells as well as FAS expression on B16-F10 melanoma to induce tumor cell apoptosis in vitro and in vivo [143]. Based on microarray gene expression data, it seems that the treatment with chidamide, a novel oral benzamide class of HDAC inhibitors, exhibits the up-regulation of cytotoxic enzymes such as granzyme H, granzyme A, and perforin 1 in peripheral white blood cells from patients with T cell lymphoma [144]. Moreover, treatment with entinostat leads to CD4 cytotoxic T lymphocyte differentiation by inducing the expression of cytotoxic genes such as *granzyme B*, *Tbx21*, and *Ifng* through upregulation of a Runx3/CBFβ-dependent pathway, which is a key factor for the development of CD8 effector T cells [145,146]. On the other hand, TSA suppresses the growth of B16-F0 melanoma through inhibition of CD4 T cell apoptosis by specifically down-regulating FASL expression on tumor-infiltrated CD4 T cells [147]. These studies suggest that T cell’s cytotoxic activity can be enhanced by HDAC inhibitors, thereby contributing to anti-tumor immunity. However, the immunomodulatory effect of HDAC inhibitors on T cell cytotoxicity against GvHD target organs has not been well elucidated. 

### 4.2. Effect of EZH2 Destabilization through HDAC6 Inhibition on GvHD and GvL

Acetylation of protein also affects protein stability by the ubiquitin-dependent proteasome degradation system. The acetylation of lysine has been shown to increase stability in these proteins: HSP90, FOXP3, p53, p73, SMAD7, SREBP1a, RUNX3, SF-1, ER81, FOXO4, NF-E4, HNF6, and E2F1. Enhancer of Zeste Homolog 2 (EZH2) is a histone-lysine N-methyltransferase 2 and highly expressed on actively dividing but not resting T cells. Genetic deletion of *Ezh2* in donor T cells or pharmacologic inhibition of EZH2 has been shown to prevent GvHD without affecting the GvL effect in mouse models of allo-HSCT [148,149]. The stability of EZH2 is regulated by HSP90 through its chaperone function that protects EZH2 from protein degradation. Deacetylation of HSP90 by HDAC6 increases chaperone activity, whereas hyperacetylation of HSP90 inhibits the activity. Hyperacetylation of HSP90 by HDAC6 inhibitor, therefore, inhibits tumor growth and survival by destabilizing proteins related to pro-growth and pro-survival oncoproteins such as BCR-ABL, AKT, and BRAF. Likewise, destabilizing EZH2 protein by HSP90 inhibitor, AUY922, reduces GvHD and tumor burden, leading to significantly improved overall survival in a mouse model of allo-HSCT [148]. In addition, panobinostat has been shown to maintain the acetylation status of HSP90, thereby blocking its functions, leading to degradation of EZH2 in AML cells. TSA can also inhibit the deacetylation of HSP90 at K271. These studies have suggested that inhibition of HDACs may not only prevent GvHD but also reduces tumor burden through rapid EZH2 degradation in both alloreactive donor T cells and tumors by loss of HSP90 chaperone function.

## 5. Clinical Aspects of HDAC Inhibitors

Currently, broad-spectrum immunosuppressive agents are being used to suppress GvHD after allo-HSCT. However, these agents also confer an increased risk of infection and tumor relapse. JAK inhibition is one of the promising therapeutic strategies in the prevention and treatment of GvHD since pre-clinical animal studies with JAK inhibitors demonstrated a significant reduction of GvHD while maintaining or enhancing immune reconstitution compared to control groups [124,125,149,150]. Consistent with these pre-clinical studies, ruxolitinib was highly effective for steroid-refractory GvHD [151,152]. Similarly, the first prospective clinical trial with a JAK inhibitor, itacitinib (a selective JAK 1 inhibitor; ClinicalTrials.gov #NCT03139604), demonstrated encouraging efficacy in acute GvHD patients [153]. Nonetheless, a phase III clinical trial with itacitinib by Incyte Corporation (Wilmington, DE, USA) failed to meet the primary endpoint with no significant positive response rate. In Appendix A, we have summarized the outcomes of clinical studies that have been completed using these drugs and HDAC inhibitors for GvHD. Compared to other treatment options, vorinostat seems to be very promising as demonstrated with relatively higher overall survival and lower GvHD and relapse rates. 

HDAC inhibitors are promising anti-cancer drugs for both solid tumors and hematologic malignancies. In addition, their efficacy in treating inflammation has been demonstrated in several inflammatory diseases [154,155]. Consequently, there has been considerable interest in translating HDAC inhibitors into clinical studies. Some HDAC inhibitors, such as vorinostat, romidepsin, and belinostat, have been approved by the FDA for clinical use. However, it remains unclear how these HDAC inhibitors should be deployed for optimal clinical benefit, considering their pleiotropic roles in diverse cellular pathways that are controlled by acetylation.

Only two HDAC inhibitors, vorinostat and panobinostat, have completed clinical trials to prevent and treat GvHD after allo-HSCT (Table 2). Vorinostat was approved by the FDA for the treatment of cutaneous T cell lymphoma in October 2006. A phase II clinical trial using vorinostat combined with tacrolimus and methotrexate (ClinicalTrials.gov #NCT02409134) was conducted for prophylaxis of neurocognitive dysfunction caused by GvHD [156]. Some of the prevalent psychosocial problems associated with GvHD patients include neurocognitive impairment, depression, anxiety, sexual dysfunction, and fatigue, especially those who receive myeloablative conditioning. Vorinostat improved neurocognitive problems in both autologous and allogeneic transplant patients. Moreover, vorinostat with standard GvHD immunoprophylaxis of tacrolimus and mycophenolate demonstrated a low incidence of severe acute GvHD after Fludarabine/Busulfan/low dose of total body irradiation (Flu/Bu2/TBI)/conditioning allo-HSCT (ClinicalTrials.gov #NCT00810602). In addition to reduced incidence of acute GvHD, vorinostat showed a low incidence of relapse in hematologic malignancies including multiple myeloma, chronic lymphocytic leukemia, and lymphoma (16%) compared to relapse in phase I/II study of maraviroc with standard GvHD prophylaxis (56%), in which the cohort of patients was similar to the former in the distribution of diagnoses [157]. Consistent with preclinical studies, inhibition of HDACs by vorinostat decreased plasma levels of TNFR1 and intracellular expression of TNF-α in peripheral blood mononuclear cells of transplant patients. There is also an increase in the number of CD4 + CD25 + CD127- Treg cells and augmented expression of FOXP3 and IDO mRNA compared with control patients after allo-HSCT [158]. More recently, vorinostat combined with tacrolimus and methotrexate on high-risk unrelated donor transplantation after myeloablative conditioning was investigated (ClinicalTrials.gov #NCT01789255 and #NCT01790568). According to blood analyses, acetylated-H3 levels were increased, whereas IL-6 and GvHD biomarkers such as soluble ST2 and Reg3a were significantly reduced in vorinostat-treated patients compared with control patients. Inhibition of IL-6 receptor by a monoclonal antibody has been demonstrated to reduce the incidence of GvHD in mice [125] and human patients [159]. Likewise, vorinostat had a reduction of acute (22% in grade 2–4 and 8% in grade 3–4) and chronic GvHD (29%) as well as relapse diagnosis of AML/MDS (19%), which resulted in increased GvHD-free survival (47%) at 1 year compared to clinical outcomes with AML patients who received tacrolimus and methotrexate-based GvHD prophylaxis and other studies (incidence of 42–49% in grade 2–4 and 12–18% in grade 3–4 in acute GvHD; 45–48% in chronic GvHD; 20–40% relapse) [155]. The phase I/II study evaluated another histone deacetylase inhibitor, panobinostat, with corticosteroids in preventing and treating acute GvHD (ClinicalTrials.gov #NCT01111526). The study showed that percentages of non-relapse mortality, relapse mortality, and GvHD-related non-relapse mortality were 12.5%, 12.5%, and 6.25%, respectively, and resulted in overall survival of 69% (11 of 16) at 1 year time frame. The overall survival in this study was higher than that in a larger study with AML patients receiving allo-HSCT [5]. These studies suggest that vorinostat and panobinostat might be promising agents to prevent and treat GvHD while enhancing GvL. However, since these two drugs are broad-spectrum HDAC inhibitors, it would be essential to develop more selective HDAC inhibitors. 

## 6. Conclusions

Despite significant progress in allo-HSCT, GvHD remains a major clinical complication. Although many therapeutic strategies have been explored to prevent GvHD, most strategies also limit GvL. Acetylation of histones and non-histone proteins often reduces tumor growth by upregulating tumor suppressor genes and anti-oncogenes as well as stabilizing proteins that induce cell cycle arrest and apoptosis. Hence, HDAC inhibitors have been well established as anti-tumor drugs for T cell lymphoma, multiple myeloma, non-small lung cancer, and so forth, with/without combination therapy such as bortezomib and dexamethasone. Recently, several HDAC inhibitors demonstrated anti-inflammatory properties and immunomodulatory effects in GvHD and several autoimmune diseases. All aspects of HDAC inhibitors in GvHD pathogenesis and GvL effects discussed above are summarized in Figure 1. Vorinostat and panobinostat have been shown to lower the incidence of GvHD and relapse significantly better than the standard GvHD prophylaxis in human patients [5,160]. In addition, vorinostat and VPA attenuate GvHD and improve leukemia-free survival in mouse models of GvHD and GvL. Although HDAC inhibitors, in general, appear to prevent GvHD pathogenesis, some contradictory results have been reported. The reason for such contradictory results may be that there are more than 3600 lysine residues in 1750 histones and non-histones whose acetylation is regulated by HDACs. Besides, it is difficult to selectively target only GvHD-causing HDACs due to the conserved catalytic site. Thus, it is conceivable that classes of HDAC inhibitors, different potencies in inhibiting each member of the HDAC family, types of target cells, types of GvHD models, and dosage and time periods of inhibitors could result in these contradictory outcomes in GvHD. 

To date, non-selective HDAC inhibitors have been tested for the prevention of GvHD without affecting GvL. However, none of them is approved for GvHD in clinical use. Therefore, further studies are needed to develop novel HDAC inhibitors that surpass the efficacy and specificity of those pan-HDAC inhibitors in preventing GvHD while preserving GvL after allo-HSCT.

## Figures and Tables

**Figure 1 ijms-21-04281-f001:**
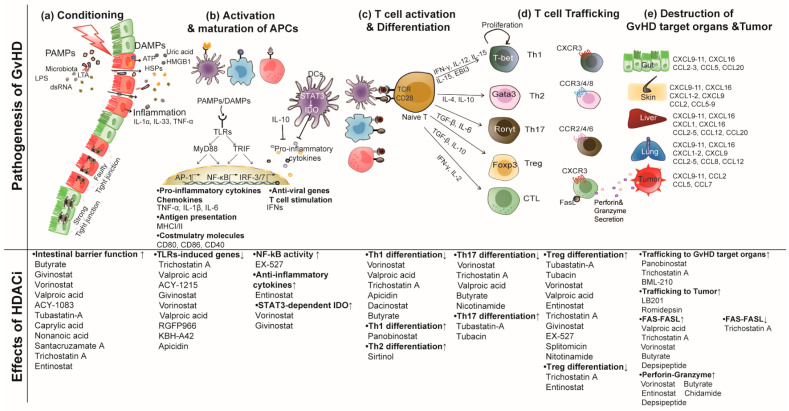
Overview of the histone deacetylases (HDAC) inhibitors on graft-versus-host disease (GvHD) pathogenesis. The pathogenesis of GvHD in sequential steps; (**a**) conditioning regimen; (**b**) activation and maturation of the antigen-presenting cells (APCs); (**c**) donor T cell activation, expansion, differentiation; (**d**) trafficking of donor T cells; (**e**) destruction of GvHD target organs. In the bottom panel, various HDAC inhibitors are listed for each stage of the GvHD pathogenesis. Some of the inhibitors such as trichostatin A and entinostat have contradictory effects on GvHD pathogenesis, particularly in the regulation of Treg cell differentiation. It is assumed that these dual effects might be derived from different types of GvHD models, and dosage and timing of inhibitors.

**Table 1 ijms-21-04281-t001:** Classification of HDACs.

Family	Class	Homologous to Yeast		Members	Localization	Size (aa)	Selective Inhibitors
Zn^2+^-dependent HDACs	I	RPD3		HDAC1	Nucleus	483	CM-675
				HDAC2	Nucleus	488	Santacruzamate
				HDAC3	Nucleus	423	RGFP966
				HDAC8	Nucleus	377	HDAC8-IN-1
	II	HDA1	a	HDAC4	Nucleus/ cytoplasm	1084	Tasquinimod
				HDAC5	Nucleus/ cytoplasm	1122	
				HDAC7	Nucleus/ cytoplasm	855	
				HDAC9	Nucleus/ cytoplasm	1011	
			b	HDAC6	Mainly cytoplasm	1212	ACY-1083/ ACY-1251/ J22352/ Tubastatin A/ Tubacin
				HDAC10	Mainly cytoplasm	669	
	IV	RPD3/HDA1		HDAC11	Nucleus/ cytoplasm	347	FT895
NAD+-dependent HDACs	III	SIR2		SIRT1	Nucleus/ cytoplasm	389	Selisistat (EX 527)
				SIRT2	Mainly cytoplasm	399	AK 7/ Thiomyristoyl/ AGK2
				SIRT3	Mitochondria	314	3-TYP
				SIRT4	Mitochondria	310	
				SIRT5	Mitochondria	355	
				SIRT6	Nucleus	400	OSS_128167
				SIRT7	Nucleus	347	97491

**Table 2 ijms-21-04281-t002:** HDAC inhibitors in the completed clinical trial in the United States.

Name	Treatment with	Disease Setting	Clinical Phase
Panobinostat (LBH589)	Glucocorticoids	Graft-Versus-Host Disease Homologous Wasting Disease	phase I/II (NCT01111526)
Vorinostat	tacrolimus, methotrexate	Graft-Versus-Host Disease Quality of Life	phase II (NCT02409134)
	tacrolimus, mycophenolate	Graft-Versus-Host Disease Hematologic Malignancies	phase II (NCT00810602)
	tacrolimus, methotrexate	Graft-Versus-Host Disease Hematologic Neoplasms Non-Neoplastic Hematologic and Lymphocytic Disorder	phase II (NCT01790568)
	tacrolimus, cyclosporine, methotrexate	Graft-Versus-Host Disease Chronic Myelogenous Leukemia Adult Acute Myeloid Leukemia Adult Lymphomatoid Granulomatosis B-cell Chronic Lymphocytic Leukemia Adult Burkitt Lymphoma Adult Diffuse Small/Large/Mixed Cell Lymphoma Adult Immunoblastic Large Cell Lymphoma Adult Lymphoblastic Lymphoma Follicular Lymphoma Mantle Cell Lymphoma Marginal Zone Lymphoma Small Lymphocytic Lymphoma Cutaneous B-cell Non-Hodgkin Lymphoma Extranodal Marginal Zone B-cell Lymphoma of Mucosa-associated Lymphoid Tissue Intraocular Lymphoma Myelodysplastic Syndrome with Isolated Del(5q) Myelodysplastic/Myeloproliferative Neoplasm, Unclassifiable Nodal Marginal Zone B-cell Lymphoma Post-transplant Lymphoproliferative Disorder Central Nervous System Hodgkin Lymphoma Central Nervous System Non-Hodgkin Lymphoma Recurrent Adult Hodgkin Lymphoma Refractory Anemia Refractory Chronic Lymphocytic Leukemia Refractory Cytopenia With Multilineage Dysplasia	phase II (NCT01789255)

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
