# Peer review of "Targeting Histone Deacetylases to Modulate Graft-Versus-Host Disease and Graft-Versus-Leukemia"

_ijms, 2020, doi:10.3390/ijms21124281_

Round 1

Reviewer 1 Report

Kim et al. in their manuscript “Targeting histone deacetylases to modulate graft-versus-host disease and graft-versus-leukemia review the data on the current clinical features of HDAC inhibitors in preventing and treating GvHD as well as maintaining GvL.

This is a very comprehensive review with the most recent data on the administration of HDAC inhibitors in the treatment of acute graft versus-host disease, which represents the major complication of allogeneic hematopoietic stem cell transplantation.  In particular, the authors focused on the current knowledge on clinical observations of HDAC inhibitors in preventing and treating GvHD while maintaining GvL. The authors clearly described pathogenesis of GvHD and the effect of HDAC inhbitors in each stage of the disease. Overall, the authors presented these data in an elegant and understandable way for the broader readership.

However, what I miss in this review, is a comparison between the clinical effects of HDAC inhibitors and other substances e.g. steroids, JAK-inhibitors, various humanized monoclonal antibodies etc. which are currently used for the treatment of aGvHD in addition to cell-based therapy such as mesenchymal stromal cells. The authors could insert these data in the section 5: Clinical aspects of HDAC inhibitors.

Author Response

Response to reviewers’ comments:

We highly appreciate the reviewer’s insightful and helpful comments on our review manuscript “Targeting histone deacetylases to modulate graft-versus-host disease and graft-versus-leukemia”. In this letter, we have addressed the points raised by the reviewers. In particular, we added Supplementary Table 1, which summarizes the outcomes of clinical trials that have been completed for the treatment of GvHD. Our response to reviewers’ comments is highlighted in yellow.

Reviewer #1: This is a very comprehensive review with the most recent data on the administration of HDAC inhibitors in the treatment of acute graft-versus-host disease, which represents the major complication of allogeneic hematopoietic stem cell transplantation.  In particular, the authors focused on the current knowledge on clinical observations of HDAC inhibitors in preventing and treating GvHD while maintaining GvL. The authors clearly described pathogenesis of GvHD and the effect of HDAC inhbitors in each stage of the disease. Overall, the authors presented these data in an elegant and understandable way for the broader readership.

However, what I miss in this review, is a comparison between the clinical effects of HDAC inhibitors and other substances e.g. steroids, JAK-inhibitors, various humanized monoclonal antibodies etc. which are currently used for the treatment of aGvHD in addition to cell-based therapy such as mesenchymal stromal cells. The authors could insert these data in the section 5: Clinical aspects of HDAC inhibitors.

Our response: We would like to thank the reviewer for bringing up the important point and valuable suggestions. In line with the reviewer’s comment, we added Supplementary Table 1. Unfortunately, there is no study that directly compares the clinical efficacies of HDAC inhibitors and other agents in human patients. Thus, we summarize the outcomes of clinical trials that are completed. Our summary includes overall survival, the incidence of acute GvHD, and relapse. Also, we have added a recent update on JAK inhibitors in clinical trials (lines 378-389).

Reviewer #2: The authors described the role of HDAC inhibitors in controlling GVHD and preserving GVL effect in this review. The manuscript is well organized and written comprehensively.

Specific comment

Since the description from lines 257 to 285 is not always related to the pathogenesis of GVHD and GVL, these explanations should be deleted.

Our response: We appreciate the encouraging comments and valuable suggestion. We agree with the reviewer that the section (originally in lines 257-285) is not always related to the pathogenesis of allo-HSCT. We believe, however, that this section might be able to provide insights into the effect of HDAC inhibitors on Treg cells and the mechanisms underlying the increased Treg cells in allo-HSCT. Thus, in this revised manuscript, right before this problematic section, we added a paragraph as follows (lines 257-263);

“Consistent with these studies, vorinostat-treated patients demonstrated an increase in the number and function of Treg cells with upregulated CD45R and CD31 that are representative of enhanced suppressive Treg cell function after allo-HSCT [82]. Whereas there is only a limited number of studies that investigated the effect of HDAC inhibitors on Treg cells in allo-HSCT, many studies have been performed outside the field of allo-HSCT, such as solid organ transplant rejection and DSS-induced colitis. Thus, these studies may provide important insights into understanding the mechanisms by which HDAC inhibitors modulate Treg cells.”

Reviewer 2 Report

   The authors described the role of HDAC inhibitors in controlling GVHD and preserving GVL effect in this review.  The manuscript is well organized and written comprehensively. 

Specific comment

Since the description from line 257 to 285 is not always related to pathogenesis of GVHD and GVL, these explanation cshould be deleted.

Author Response

(The authors gave the same response as above.)
